# Male Infertility is a Women’s Health Issue—Research and Clinical Evaluation of Male Infertility Is Needed

**DOI:** 10.3390/cells9040990

**Published:** 2020-04-16

**Authors:** Katerina A. Turner, Amarnath Rambhatla, Samantha Schon, Ashok Agarwal, Stephen A. Krawetz, James M. Dupree, Tomer Avidor-Reiss

**Affiliations:** 1Department of Biological Sciences, University of Toledo, Toledo, OH 43606, USA; Katerina.Turner@utoledo.edu; 2Department of Urology, Vattikuti Urology Institute, Henry Ford Health System, Detroit, MI 48202, USA; arambha1@hfhs.org; 3Division of Reproductive Endocrinology & Infertility, Department of Obstetrics and Gynecology, University of Michigan Medical School, L4000 UH-South, 1500 E. Medical Center Drive, Ann Arbor, MI 48109, USA; sschon@med.umich.edu; 4American Center for Reproductive Medicine, Cleveland Clinic, Cleveland, OH 44195, USA; agarwaa@ccf.org; 5Department of Obstetrics and Gynecology, Center for Molecular Medicine and Genetics, C.S. Mott Center for Human Growth and Development, Wayne State University School of Medicine, Detroit, MI 48201, USA; steve@compbio.med.wayne.edu; 6Department of Urology and Department of Obstetrics and Gynecology, University of Michigan, Ann Arbor, MI 48019, USA; jmdupree@med.umich.edu; 7Department of Urology, College of Medicine and Life Sciences, University of Toledo, Toledo, OH 43614, USA

**Keywords:** male fertility, sperm, semen analysis, centriole: RNA, oxidative stress, DNA fragmentation, women’s health

## Abstract

Infertility is a devastating experience for both partners as they try to conceive. Historically, when a couple could not conceive, the woman has carried the stigma of infertility; however, men and women are just as likely to contribute to the couple’s infertility. With the development of assisted reproductive technology (ART), the treatment burden for male and unexplained infertility has fallen mainly on women. Equalizing this burden requires reviving research on male infertility to both improve treatment options and enable natural conception. Despite many scientific efforts, infertility in men due to sperm dysfunction is mainly diagnosed by a semen analysis. The semen analysis is limited as it only examines general sperm properties such as concentration, motility, and morphology. A diagnosis of male infertility rarely includes an assessment of internal sperm components such as DNA, which is well documented to have an impact on infertility, or other components such as RNA and centrioles, which are beginning to be adopted. Assessment of these components is not typically included in current diagnostic testing because available treatments are limited. Recent research has expanded our understanding of sperm biology and suggests that these components may also contribute to the failure to achieve pregnancy. Understanding the sperm’s internal components, and how they contribute to male infertility, would provide avenues for new therapies that are based on treating men directly for male infertility, which may enable less invasive treatments and even natural conception.

## 1. Introduction

The contribution of men and women to reproduction is uneven. While men provide the sperm and women provide the egg, women must also carry and deliver the pregnancy. Therefore, it is a biological inevitability that women are physically burdened more than men by fertility treatments. However, a woman undergoing extensive fertility treatments for male infertility is not a biological inevitability. Instead, we argue that deficiencies in our understanding of the biology of male fertility and the common practice of solely employing traditional semen analysis to diagnose male infertility have led to an uneven distribution of treatment. This situation is largely due to the success and availability of assisted reproductive technology (ART), which disproportionately burdens women. The perception that ART has “solved” male infertility has hindered male infertility research, development of novel diagnostics, and the advancement of treatments that would either improve treatment options or help couples achieve natural conception [1].

In this paper, we suggest that basic and translational research of sperm biology is needed to increase our knowledge of male infertility and find new treatments to optimize natural conception and ART outcomes. Currently, the absence of andrologists from the infertility treatment team often leads to the lack of male infertility evaluation and the premature adoption of ART as the first and only treatment option. This inherently means women are the patients when male infertility is the disease. Identifying new sperm components that can be pharmacologically or otherwise corrected may provide less invasive or expensive pathways to remediation of infertility in the future. Additionally, in cases of unexplained infertility, examining these components can point to the man as the main contributor to the couple’s infertility, which may be critical when the woman has a low ovarian reserve.

This manuscript is intended to prompt discussion and investigation—it is not intended as a guideline—but is instead aimed to push the field of male infertility research forward. We start this discussion by describing how men are as likely as women to be responsible for infertility. We then describe the arduous treatments women undergo when men are the reason for the couples’ infertility. Subsequently, we explain how the deficiencies in traditional semen analysis additionally burdens women. Furthermore, we propose that there are undiagnosed and potentially treatable sperm defects that could be addressed by exploring advanced and experimental sperm testing. Finally, we argue that stimulating research in male infertility will lead to the development of new diagnostic techniques and new treatments that more directly address male infertility. The revival of this research will ultimately benefit women’s health and help to ensure the birth of a healthy child. This argument is presented in the context of several sperm components that are under investigation, possible new diagnostic tools that clinicians may employ, the need for multi-parametric analysis, and potential new treatments that would directly address the causes of male infertility.

Of note, in this manuscript, we focus on the sperm’s role in male infertility; however, similar arguments can be made about other contributing elements. For example, the male genital tract [2], seminal fluids [3,4], hormonal regulation [5,6], non-germ line testicular cells [7,8], and his genetic constitution can all contribute to male infertility [9]. Finally, we focus on how investing in the identification of new male infertility mechanisms will improve the treatment outcomes and lives of women in the future.

## 2. Men Are as Likely as Women to Be Responsible for Infertility

Infertility is quite common, with approximately one out of eight couples reporting the inability to conceive after one year of trying [10,11,12]. Through the use of traditional semen analysis and other classic diagnostic tests, approximately 1/3 of infertility is attributed to male factors, 1/3 attributed to female factors, and in 1/3 of couples no cause can be identified, also known as unexplained or idiopathic infertility (Figure 1) [13,14]. Some of the unknown causes of infertility may be due to obscure disorders in the man, the woman, or the couple (i.e., male and female factors that, when combined, result in that couple’s infertility) [15,16]. Therefore, in general, men are as likely as women to be responsible for infertility.

Additionally, it is important to note that men with a normal semen analysis merit additional diagnostics. This is because semen analysis does not provide information on the state of the internal sperm contents, which can be a source of unexplained infertility. For example, some studies report that up to 40% of men from couples with recurrent pregnancy loss had normal sperm density and motility, but had sperm aneuploidy [17]. While the role and degree of sperm DNA defects are controversial [18], there are examples where, even with normal semen analysis, men can have sperm with abnormal components.

## 3. Current Infertility Treatments Disproportionality Burden Women

Due to the lack of direct treatments for men, many treatments for male infertility involve treating the woman; ideally, male infertility treatments would improve sperm quantity or quality, and possibly the ability to conceive naturally (Figure 1). For example, the woman may carry most of the treatment burden when the man has a low count, reduced motility, or has abnormal sperm morphology [19]. Similarly, when a couple is infertile, the sperm may appear normal, but be dysfunctional due to an obscure cause. In this case, the woman may be treated using ART, which disproportionately physically burdens the woman. Furthermore, even if a potential cause of the woman’s infertility is identified, it does not preclude the man from also having a fertility problem that is contributing to the couple’s infertility [20].

When a couple is unsuccessful with pregnancy it is common for the female partner to be the first to seek medical advice [21]. The initial evaluation of the infertile female patient involves an extensive history and diagnostic assessment, including multiple laboratory tests, ultrasound imaging, and a hysterosalpingogram [22]. Conversely, the evaluation and diagnosis of male patients is far less thorough and far less invasive.

If the diagnosis is male infertility or an unknown cause of infertility, the main treatments available disproportionately affect women and usually include ART treatments such as intrauterine insemination (IUI), in vitro fertilization (IVF), or intracytoplasmic sperm injection (ICSI) and its variants.

Timed intercourse (TIC) is the least invasive intervention and is normally only used when the man has a normal sperm count and the woman appears normal or has an ovulatory defect, such as polycystic ovarian syndrome (PCOS), which can be managed. Treatment involves determining the window of ovulation and planning intercourse during that timeframe to maximize the chance of achieving pregnancy. This treatment can be equally stressful on the couple because both partners need to perform on-demand [23].

Intrauterine insemination (IUI) is commonly the first major intervention for women when the couple is having trouble conceiving, especially for couples with unexplained infertility and mild male infertility [24]. IUI is the direct deposition of sperm into the uterus with a catheter that is threaded through the cervix and into the uterus, often after the woman has taken hormonal medications to stimulate the ovaries. These hormones have many potential side effects such as hot flashes, headache, nausea, and an increased chance of multiple gestations. While, IUI is physically uncomfortable, it is relatively non-invasive, and is less expensive than other forms of ART.

In vitro fertilization (IVF) and intracytoplasmic sperm injection (ICSI) are the most invasive interventions. IVF involves the fertilization of an egg (oocyte) in a laboratory dish outside of the body, while ICSI is the direct injection of a sperm cell into the oocyte. In both IVF and ICSI, the embryo is placed back into the uterus where it can hopefully result in a healthy pregnancy. Both interventions involve hormonal stimulation and oocyte retrieval from the female. There is a potential for serious side effects from medications or during the actual oocyte retrieval process such as ovarian hyperstimulation syndrome (OHSS), which, while rare, can lead to significant morbidity [25]. These treatments are a clear case of a greater burden being placed on the woman rather than the man when she is potentially not the infertile partner. Additionally, these treatments cannot overcome sperm with defective interior components. Indeed, in one study, infertile couples that opted to use artificial insemination with donor sperm had a pregnancy rate of 44%. When infertile couples elected to use intra-couple sperm for ICSI the pregnancy rate per cycle was only 12% [26].

The first human born after the successful use of ICSI occurred on January 14, 1992 [27,28], and it was thought that this procedure would solve most causes of male infertility. As a result, since then, there has been little advancement in male infertility treatments. It appears that the successful use of the above-mentioned treatments has hindered the research and development of treatment options that focus on men as an integral part of the couple.

Currently available ART necessitates a female focused treatment to optimize outcomes [29,30]. While ART can be utilized to help overcome some male factor infertility, such as low sperm concentration, it cannot overcome defective interior components of sperm (Figure 1). While the success rate of ART has significantly improved over the years, it is a complicated process and there is never a guarantee this process will result in a healthy pregnancy [31,32,33]. Indeed, the live birth rate after ART is about 30% and decreases with advanced maternal age [34]. It may take several cycles of IVF/ICSI to become pregnant, which can be very expensive, especially because the costs of these treatments are frequently borne by the patient [35]. As discussed, these treatments are associated with increased severe maternal morbidity (6.0% for pregnancies conceived via IVF versus 2.1% for natural conception pregnancies), and would be wasteful if the cause of infertility is due to an internal defect in the sperm [36]. Therefore, it is beneficial to know as much as possible about both female and male causes of infertility so that the couple can be best counseled.

Currently, many treatments for male infertility involve ICSI or IVF. An improvement to this current treatment would be a male treatment that improves sperm quality enough to use IUI or TIC instead of ICSI or IVF. However, the ultimate goal of treating male infertility is that it will lead to natural conception. In the case of unexplained infertility, the goal is to be able to make a conclusive decision on the contribution of male to infertility so that treatment can be directed correctly.

## 4. There Are Many Barriers to Male Infertility Diagnosis and Treatment

There are multiple epidemiologic, geographic, epistemic, financial, socioeconomic, and policy-based barriers that, compounded with our lack of knowledge, make it challenging for men to obtain high-quality infertility care [37]. One barrier to men seeking infertility treatment is that male specialists (andrologists) are not usually part of the infertility treatment team. However, even if this were to change, there would be an insufficient number of andrologists in the United States, where there are only approximately 200 andrologists [38]. This makes general urologists or even obstetrician/gynecologists (OB/GYN) the main source of care for infertile men [39]. For historical reasons, in some European countries, physicians such as dermatologists, and endocrinologists can act as andrologists [40]. The lack of andrologists is especially concerning because there is a trend of decreasing male fertility [41,42,43]. A systemic review by Levine et al. found that sperm counts decreased by approximately 50% between 1973 and 2011 [44].

Unfortunately, it is often thought that male causes for infertility can be overcome with ART, even without a complete male andrological evaluation. Consequently, 18–27% of the time, in a couple presenting with infertility, the man does not undergo any infertility evaluation [45,46]. This bias in diagnosis could lead to lengthening the time of treatment, undermining women’s health, and ability to achieve the birth of a healthy child. Ideally, the couple should be seen together to have their history taken and be examined by male and female reproductive experts, respectively [47]. By treating the couple in parallel, missed information about the couple’s health that could be contributing to infertility could be reduced. This can prevent the premature and unnecessary use of ART [48]. Both partners can support each other through the stress they feel when each understands the conditions that confound their inability to conceive [10,49,50,51,52]. Knowing they are not alone in their treatment and are doing all they can reduces the burden of infertility on both the man and the woman [49].

## 5. Traditional Semen Analysis Only Tests General Sperm Properties, Which Are Not Necessarily Predictive of Fertility

It is well established that the main role of sperm is to travel to the oocyte. Of the millions of sperm in the ejaculate, only a few hundred will reach the oocyte with just one ultimately fertilizing the egg [53,54]. Therefore, traditional semen analysis has focused on the journey of the sperm. However, traditional semen analysis has several limitations that prevent it from being fully able to predict a sperm’s ability to travel and reach the oocyte [55]. These limitations are partially because the reference values for semen analysis are based on information gathered solely from men in fertile couples [56]. The impact of the parameters tested on infertility is not well known, and normal semen analysis does not necessarily predict successful treatment. Furthermore, sperm counts and motility are highly variable between patients, making it difficult to give clear cut-off values to predict a man’s fertility potential. Lastly, many of the tests performed are subjective leading to variability between laboratories and individuals interpreting the results, which can lead to differences in a male patient’s care [57]. Therefore, men who have a normal semen analysis may still have deficient sperm that cannot reach the oocyte, fertilize the oocyte, or contribute to a healthy pregnancy.

## 6. There Are Undiagnosed and Potentially Treatable Sperm Defects

The presence of sperm components that are not well studied indicates that there are undiagnosed, but potentially treatable, causes of infertility [58,59]. These components include DNA, RNA, proteins, and sperm structures, such as the centrioles (Figure 2). With adequate investment in basic and translational research on these components, a determination of the extent to which these components contribute to male infertility can be made. We can divide the research efforts into two types based on the stage of the research to reveal the role of these sperm components. In the first type, the research is in an advanced translational mode, although there is a lack of a definitive, conclusively, and generally accepted documentation of an impact on infertility—advanced sperm testing. In the second type, the research is at the basic research stage or early translational mode, and much more work needs to be carried out to obtain definitive documentation of an impact on infertility—experimental sperm tests.

## 7. Advanced Sperm Tests Examine Sperm DNA

The traveling sperm contributes several internal components that are essential for embryo development and pregnancy. One of these internal components is DNA, which is the genetic code. DNA is usually examined when other tests fail to resolve a cause for the infertility [60,61]. Currently, DNA is being evaluated in clinics using two types of tests. These include determining if there are deletions in Y chromosome genes (y-micro deletion) and examining the visual appearance or number of chromosomes by traditional karyotyping or spectral karyotyping). These tests only consider some potential abnormalities in sperm DNA. In addition to the currently used DNA tests, two other advanced tests analyze DNA: DNA fragmentation and oxidative stress.

DNA fragmentation assessment can benefit couples that have not had success with previous IVF/ICSI cycles or have had repeated miscarriages. If the DNA fragmentation index is elevated, there are several potential treatments. These treatments include follicle-stimulating hormone (FSH) [1], magnetic assisted cell sorting (MACS) that may help select sperm with high-quality DNA [62], and retrieval of testicular sperm (known as testicular sperm extraction, TESE, or testicular sperm aspiration, TESA) for use with ICSI [63,64]. The diagnosis and treatments for DNA fragmentation have not been fully validated using randomized control trials. Future investigations with demonstrated sensitivity and specificity are required.

Oxidative stress (recently referred to as Male Oxidative Stress Infertility, MOSI) is thought to be responsible for 30–80% of male infertility [65,66]. When increased sperm oxidative stress is suspected, lifestyle modifications and oral antioxidant therapy are often employed, as it is relatively inexpensive, readily available, and easily administered, with minimal side effects. However, beneficial effects vary [67]. While DNA fragmentation and oxidative stress tests are newer, they are becoming more mainstream after being validated by recent studies [68,69,70,71]. However, there is still some controversy about whether treating oxidative stress is beneficial [72].

Altogether, current and advanced sperm testing of DNA provides the physician with important, detailed information on some aspects of the internal health and functionality of the sperm.

## 8. Experimental Sperm Tests Aim to Assess Newly Discovered Sperm Components

Because we now realize the importance of sperm health, diagnostic experimental tests focusing on recently discovered essential sperm components are being designed and investigated by scientists (Figure 2).

### 8.1. RNA—Sperm Transcripts

RNAs (RNA elements and messenger RNA) are found in the sperm head, nucleus, and residual cytoplasm of the sperm [73,74,75]. Sperm RNA is implicated in regulating the epigenetic code of the embryo [76]. There are no treatments currently available for defects in the RNAs that are reflective of disease, but it does provide a mechanistic understanding of the basis of infertility [77,78,79]. RNA content may be a predictor of human health and could be used to monitor the impact and recovery from environmental exposures [80,81]. As a result, assessing sperm RNA has the potential to evaluate the father’s contribution, his current health status, and the health of his future child [80].

### 8.2. Proteins—Sperm Proteome

Sperm are transcriptionally and translationally inert cells that are dependent on already existing proteins. Hence, identifying the sperm proteome using mass spectroscopy may play a crucial role in determining a man’s fertility potential [82]. Aberrant expression of sperm proteins was reported to affect the molecular mechanisms associated with motility, capacitation, acrosomal reaction, and sperm-oocyte interaction in unexplained male infertility cases [83]. For example, the proteins involved in chromatin assembly were defective in unexplained male infertility patients [84,85,86]. However, these findings require clinical validation. Identifying deficiencies in the expression or function of any protein may be resolved by introducing the protein itself, its DNA, or RNA during ICSI.

### 8.3. Activation Factors

Oocyte activation factors, such as phospholipase C zeta (PLCζ), are a type of sperm protein that are critical to initiating embryo development [87,88]. Deficiencies in PLCζ can be treated with Ca^2+^, which increases ICSI treatment outcome [89]. ICSI can also be supplemented with recombinant human PLCζ protein, which can help to activate mouse oocytes and foster development to the blastocyst stage [90].

### 8.4. Centrioles

Sperm have two centrioles, which are large protein-based structures, located in their neck [91]. Centrioles are important for motility and normal embryo growth, [92,93,94,95]. Currently, there are no clinically approved diagnostic tests or treatments for centriole-based defects, but there is a hypothetical treatment. It may be possible to treat the infertility by combining the DNA from the father encapsulated in the sperm head with sperm centrioles from the sperm tail of a donor; a similar procedure to mitochondrial donation for females [96,97]

### 8.5. Multi-Parametric Analysis

Despite best efforts, none of the current male infertility assessments are enough to precisely predict sperm quality, and the search for a single assay that could predict fertility has been mostly fruitless [98,99,100]. The assessment deficiency is echoed by the presence of a large population of unexplained infertility, as well as the low ability of each individual assay to predict a successful pregnancy [101,102,103]. While many of these assays currently available or under development have shown correlations with fertility, in most cases they have been highly variable depending on the study [104,105]. Therefore, it appears that multi-parametric analysis of semen is needed [106]. Multi-parametric analysis of semen can be achieved using computer-aided sperm analysis (CASA) systems, but they focus on the general properties identified by semen analysis [107]. However, up to now, multi-parametric analysis of sperm components by simultaneously testing multiple independent mechanisms has not been done.

## 9. The Clinical Benefit of Identifying Male Infertility Due to Internal Sperm Components

Health professionals have several goals when identifying male infertility with the use of standard, advanced, or experimental diagnostics of sperm [108]. The first: identifying any treatable causes such as oxidative stress and DNA fragmentation. The second: identify any irreversible conditions that could still be treated with ART. The third: diagnose any currently untreatable causes of male infertility, which may include defects in the RNA and centrioles. In this last case, understanding the underlying cause of male infertility may change the course of action for couples that have had several rounds of failed ART treatment, and it may help the couple to decide to use donor sperm. A similar situation, where the couple may want to use a sperm donor, is when the woman has a low ovarian reserve and time is a critical factor; identifying a defect in the sperm’s internal components and shifting to donor sperm may help the couple to become parents before they can no longer provide the oocyte. Another option is fertility preservation where the woman freezes her eggs to give clinicians more time to correct the male infertility.

A future diagnostic test may be helpful even if there are no obvious treatment strategies because it may predict whether the use of ART will be successful. This information could then be used to counsel couples in their decision on whether to proceed with possibly expensive and futile treatments.

Identifying male infertility in infertile couples can have two benefits. In the short term, it can direct treatment efforts, e.g., increase sperm count to a range where IUI of TIC may be an option instead of IVF/ICSI. These efforts are beneficial because both IUI and TIC are less invasive and offer less expensive treatment than IVF/ICSI. In the long term, it can direct research to allow for the discoveries of treatments that improve male infertility to a point where it enables natural conception.

## 10. Conclusions

“Are sperm tests other than the WHO parameters useful in evaluating male fertility?” is a top priority question according to An International Priority Setting Partnership for Infertility https://www.focusonreproduction.eu/article/ESHRE-Meetings-research-gaps-19. Furthermore, understanding the male side of infertility was recognized as one of the top immediate projects that could be undertaken to help best ensure a successful pregnancy and the birth of a healthy child in the Michigan Action Plan for Fertility and Assisted Reproductive Technology (https://www.michigan.gov/documents/mdhhs/MiART_FullReport_3.2.18_COMBINED_616266_7.pdf). Both the American Society for Reproductive Medicine (ASRM) and the National Institute for Health and Clinical Excellence (NICE, United Kingdom) recommend that a full andrological examination of the male partner by a qualified andrologist is essential to identify male infertility and its contribution to the couple infertility [22] http://guidance.nice.org.uk/cg156.

Semen analysis is only the first step, and may fail to identify male infertility because it does not analyze many of the cell biology-based potential causes of infertility. Similarly, DNA fidelity, only one of many cell biology-based potential causes of infertility, can be analyzed, but is not regularly evaluated. Many other sperm components are important for fertility, such as such as RNA, proteins, or centrioles, which need to be investigated using a multi-parametric analysis so their diagnostic potential can be realized and treatments discovered.

Semen analysis, combined with a molecular understanding of sperm function, should be able to point to the pathways and components that are involved in infertility [79,109]. The understanding afforded by some of the advanced sperm tests that are not routinely used, but should be, and others that are in development, could point to the molecular cause of many cases of male infertility. Because of our limited understanding of the male contribution, women endure the health risks of treatments for a disease that may not be theirs. Infertility must be seen as a problem that affects couples, not as an individual failure of the female to conceive. Without further investigation of male infertility by both clinicians and researchers, this stigma will persist. Indeed, future research and treatments should be aimed at correcting the underlying cause of male infertility, thereby minimizing the risks of treatment and even allowing for natural conception that enables the birth of a healthy child.

## Figures and Tables

**Figure 1 cells-09-00990-f001:**
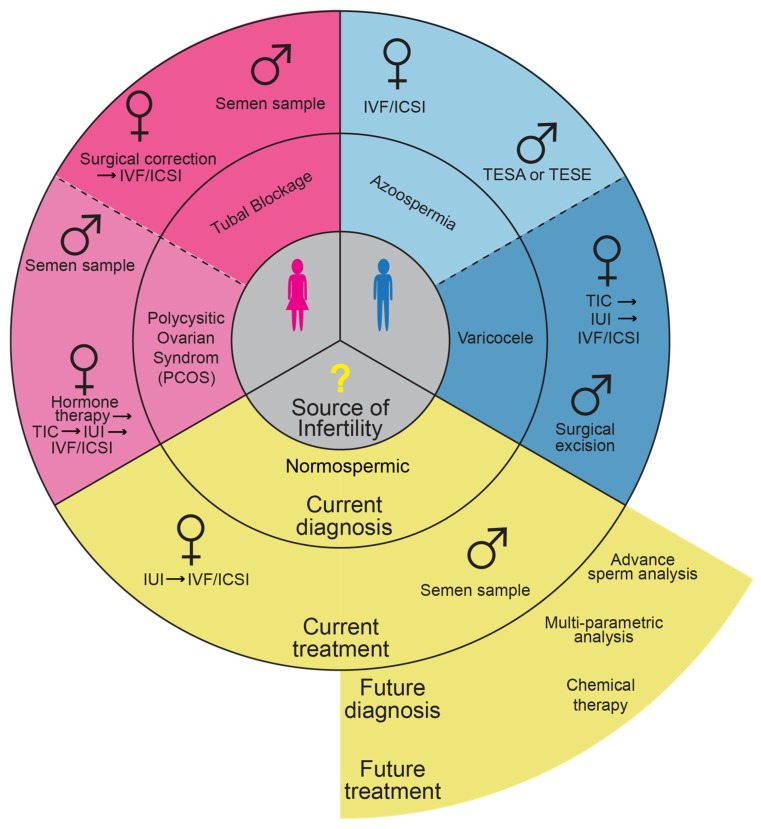
Examples of potential diagnoses and treatments that are biased against women. Male factor is indicated in blue; female factor is indicated in pink; unknown factor is indicated in yellow. For each of these factors we provide two examples of current diagnoses and their respective current treatment. In the case of unexplained infertility with obscure male cause, we also provide what should be the aim of future diagnostic tools and treatments. TESA, testicular sperm aspiration; TESE, testicular sperm extraction; TIC, timed intercourse; IUI, intrauterine insemination; IVF, in vitro fertilization; ICSI, intracytoplasmic sperm injection.

**Figure 2 cells-09-00990-f002:**
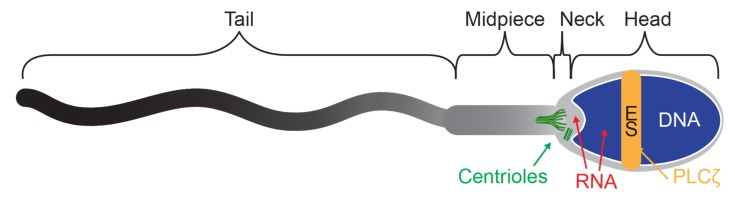
Various components of the sperm that may affect fertility. A mature sperm with DNA in the nucleus (blue), Activation protein phospholipase C zeta (PLCζ) in the equatorial segment (orange, ES), RNA throughout the sperm (red), and centrioles in the sperm neck (green).

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
