# Peer review of "Male Infertility is a Women’s Health Issue—Research and Clinical Evaluation of Male Infertility Is Needed"

_cells, 2020, doi:10.3390/cells9040990_

Round 1

Reviewer 1 Report

I evaluated the manuscript cells-742897

In my opinion, the manuscript should be deeply revised (see comments to the author,1, 8, 16-20) . The manuscript should reflect the necessity of a careful study of the male factor, often not considered in infertility clinics, as well as the necessity of an increasing research on male infertility and new diagnostic an therapeutic tools. In the present form, the manuscript seems in its first part, a ranting against an excessive burden on women health for fertility aims. It shuold be underlined by the authors that a great problem in infertility clinics is that an andrologist is often not part of the “infertility team” and/or it is not consulted, and that gynecologists often “push” the couple through ART instead of increasing diagnostic and possible therapeutic teratments before ART to obtain, as far as possible, a natural pregnancy or a pregnancy obtained with a minimal invasiveness. Too many times the diagnostic investigation is limited since ART is available, and often ART is proposed as the first option with no careful investigation of the infertile couple.

Title

1.“Male Infertility is a Women’s Health Issue”. The title seems to suggest a review on the burden of infertility and ART on women and related psychological problems. In my opinion, the title should reflect the necessity of a careful study of the male factor, often not considered in infertility clinics, as well as the necessity of more research on male infertility and new research on diagnostic and therapeutic tools. The title coul be something sounding like “Warning about the need to evaluate the male partner more in infertility clinics and to increase the investigation of the male factor”.

Abstract and Introduction

2.Line 23: “when a couple cannot conceive”: write “could not conceive”

3.Line 24: “however, men are just as likely as women to contribute to a couple’s infertility”, please rephrase, it is not clear.

4.Line 27: “infertility in men is mainly diagnosed by semen analysis”. I do not agree. Nowadays there are many scientific efforts, mainly in genetics (see Krausz et al., Expert Rev Mol Diagn. 2018 Apr;18(4):331-346), sperm biology (see, for example, Muratori M, Fertil Steril. 2015 Sep;104(3):582-90.e4.) and ultrasound (see Lotti and Maggi, Hum Reprod Update. 2015 Jan-Feb;21(1):56-83 ). Please rephrase as “Despite many scientific efforts, mainly in genetics (cit), sperm biology (cit), and male genital tract ultrasound (cit), in several infertility clinics, infertility in men is mainly diagnosed by semen analysis”. Please add the aforementioned references.

5.Line 29:  “It does not include assessment of internal sperm components such as the DNA, RNA, or organelles like centrioles, and may not necessarily reflect the health of the sperm”. Please report here only techniques with a documented impact on couple fertility, for example sperm DNA fragmentation, that has become highly requested in infertility clinics, but not RNA (strong evidence?) or organelles like centrioles.

6.Line 21: “there is no treatment available”. It is not really true. For sperm DNA framentation, a therapy with FSH has been suggested (see and add as reference Simoni et al., Hum Reprod. 2016 Sep;31(9):1960-9.), as well as with antioxidants (even if debated), and some technologies, such as MACS can be used to select the “best” spermatozoa (even id debated).

7.Introduction: see the comments above.

8.Line 51. I do not agree with this point of view “In this paper, we suggest that further research of the internal components of the sperm is necessary to distribute the burden of infertility between the partners”. The aim of this study should be not to “distribute the burden”. Our aim as researchers is to find out new diagnostic and therapeutic options in order to increase our knowledge on male infertility and find new teratments to optimize natural and ART fecundity.

9.Accordingly, please rephrase also “Identifying male components would shift the emphasis from treating women for male infertility to treating men for male infertility”.

10.Line 61. “the need for advanced and experimental sperm testing and male treatments to male infertility to alleviate this treatment inequality”. Again, this comment is dealing with sex inequality, but this is not the point (see the general comment and comment #16). Please rephrase.

Main text

11.Line 70. Please update reference 1 (see Lotti and Maggi. Nat Rev Urol. 2018 May;15(5):287-307)

12.Line 74, Figure 1. Please report some other important correctable causes of male infertility, such as hypogonadotropic hypogonadism (treatment with FSH and/or hCG) (add a reference), oligo- and/or astheno and /or teratospermia (add a reference), midline prostatic cyst and obstructive infertility.

13.Line 78. “semen analysis semen analysis” correct this mistake.

14.Line 80. Be very careful with this comment “For example, 40% of men from couples with recurrent pregnancy loss had normal sperm density and motility, but had abnormal sperm aneuploidy”. It seems that 40% of cases of miscarriages depend on male sperm aneuploidy, but it is not the case. The demonstrated male contribution to miscarriages is very low.

15.Line 97. “she may still feel responsible for being unable to conceive”. Please don’t be so definitive. There is a large literature about male negative feeling related to infertility (see and add as references Bechoua et al., Andrology 2016:4;395-403, Lotti and Maggi. Nat Rev Urol. 2018 May;15(5):287-307; Lotti et al., Hum Reprod. 2016 Dec;31(12):2668-2680.)

16.Line 157. “One barrier to men seeking infertility treatment is the lack of specialists (andrologists)”. I do not agree with this comment. Have andrologists in Europe or United States a long waiting list? I don’t think so. The great barrier for men to seek infertility treatment is that most of the infertility centers do not request or include an andrological evaluation. This is the problem: this mentality could be changed. And this mentality is mainly derived by the fact that often infertility centers are guided by gynecologists only, with no andrologists or urologiststs in their team. So far, infertility centers tend to “push” the couple to ART, often avoiding the male infertiliy investigation and research fo natural pregnancy or pregnancy with IUI. Maybe, sometimes, also economic reasons underline these decisions. Several cases where natural pregnacy could be obtained treating the male partner or with female antiestrogens treatment, and IUI are often shifted to ART with the information (and sometimes the excuse) that that techniques will allow a higher success probability. Obviously, a different scenario is when the female partner is old, for example around 40 years old, when ART is necessary, especially after longlasting infertility, to avoid a repentine reduction of ovocite quality in the next few years. However, after female 43-45 years old, ART or natural sexual intercorses have similar chances to get pregnancy.

17.Regarding DNA fragmentation, please add a paragraph on brighter and dimmer subpopulations (see and add as references Marchiani et al., Andrology. 2014 May;2(3):394-401; Muratori M, Fertil Steril. 2015 Sep;104(3):582-90.e4.) Please add also a discussion about MACS and similar technologies to select the “best” spermatozoa, and evidences for their use in clinical practice.

18.Please add a paragraph on physical (for example sexual) and psychological problems of the male partner due to infertility, and add a brief discussion on male infertility as a mirror of a poor general health (see and add as reference “Lotti and Maggi. Nat Rev Urol. 2018 May;15(5):287-307.”) Hence, underline the necessity to increase the evaluation of the male partner in infertility clinic, not only for fertility outcomes but also for investigating and treat general, sexual and psychological problems.

19.Please add a paragraph considering the utility and increasing reseasch of male genital tract ultrasound for male infertility (see Lotti and Maggi, Hum Reprod Update. 2015 Jan-Feb;21(1):56-83)

20.Please add a paragraph considering the utility and increasing reseasch of genetics on male infertility (see Krausz et al., Expert Rev Mol Diagn. 2018 Apr;18(4):331-346).

21.The manuscript should be reduced in some redundants parts.

22.A more “scientific” English or American language is required.

Author Response

Review 1

In my opinion, the manuscript should be deeply revised (see comments to the author,1, 8, 16-20). The manuscript should reflect the necessity of a careful study of the male factor, often not considered in infertility clinics, as well as the necessity of an increasing research on male infertility and new diagnostic an therapeutic tools. In the present form, the manuscript seems in its first part, a ranting against an excessive burden on women health for fertility aims. It shuold be underlined by the authors that a great problem in infertility clinics is that an andrologist is often not part of the “infertility team” and/or it is not consulted, and that gynecologists often “push” the couple through ART instead of increasing diagnostic and possible therapeutic teratments before ART to obtain, as far as possible, a natural pregnancy or a pregnancy obtained with a minimal invasiveness. Too many times the diagnostic investigation is limited since ART is available, and often ART is proposed as the first option with no careful investigation of the infertile couple. 

Answer: We would like to thank the reviewer for the many valuable comments. We believe these comments help us to strengthen the paper.

Also, we would like to thank the reviewer for making this emphasis listed below which we added to the introduction:

  1. The necessity of a careful study of the male factor that is often not considered in infertility clinics.
  2. The necessity of increasing research on male infertility, new diagnostic, and therapeutic tools. 
  3. A great problem in infertility clinics is that an andrologist is often not part of the “infertility team” and/or it is not consulted..
  4. Gynecologists are often in a position where no andrologist is nearby, and therefore  “Currently, the absence of andrologists from the infertility treatment team often leads to the lack of male infertility evaluation and the premature adoption of ART as the first and only treatment option.”
  5. Doctors need to increase diagnostic and possible therapeutic treatments before ART to obtain, as far as possible, a natural pregnancy or a pregnancy obtained with a minimal invasiveness.
  6. Too many times the diagnostic investigation is limited since ART is available, and often ART is proposed as the first option with no careful investigation of the infertile couple.  

Title

1.“Male Infertility is a Women’s Health Issue”. The title seems to suggest a review on the burden of infertility and ART on women and related psychological problems. In my opinion, the title should reflect the necessity of a careful study of the male factor, often not considered in infertility clinics, as well as the necessity of more research on male infertility and new research on diagnostic and therapeutic tools. The title coul be something sounding like “Warning about the need to evaluate the male partner more in infertility clinics and to increase the investigation of the male factor”.

Answer: We appreciate the emphasis the reviewer suggests and now change the title to include this point. The new title is: “Male Infertility is a Women’s Health Issue - Research and Clinical Evaluation of Male Infertility is Needed”

Abstract and Introduction

  1. Line 23: “when a couple cannot conceive”: write “could not conceive”

Answer: Thx - changed

  1. Line 24: “however, men are just as likely as women to contribute to a couple’s infertility”, please rephrase, it is not clear.

Answer: Thank you for pointing this out. We rephrased to: ”however, men and women are just as likely to contribute to the couple’s infertility.

  1. Line 27: “infertility in men is mainly diagnosed by semen analysis”. I do not agree. Nowadays there are many scientific efforts, mainly in genetics (see Krausz et al., Expert Rev Mol Diagn. 2018 Apr;18(4):331-346), sperm biology (see, for example, Muratori M, Fertil Steril. 2015 Sep;104(3):582-90.e4.) and ultrasound (see Lotti and Maggi, Hum Reprod Update. 2015 Jan-Feb;21(1):56-83 ). Please rephrase as “Despite many scientific efforts, mainly in genetics (cit), sperm biology (cit), and male genital tract ultrasound (cit), in several infertility clinics, infertility in men is mainly diagnosed by semen analysis”. Please add the aforementioned references.

Answer: Thank you very much for your comment. We have changed the abstract sentence to  “Despite many scientific efforts, infertility in men due to sperm dysfunction is mainly diagnosed by a semen analysis.” We also added a section in the introduction stating ”Of note, in this manuscript, we focus on the sperm’s role in male infertility; however, similar arguments can be made about other male elements. For example, the male genital tract [2], seminal fluids [reviewed in 3,4], hormonal regulation [5,6], non-germ line testicular cells [7,8], and his genetic constitution can all contribute to male infertility [9].”

  1. Line 29:  “It does not include assessment of internal sperm components such as the DNA, RNA, or organelles like centrioles, and may not necessarily reflect the health of the sperm”. Please report here only techniques with a documented impact on couple fertility, for example sperm DNA fragmentation, that has become highly requested in infertility clinics, but not RNA (strong evidence?) or organelles like centrioles.

Answer: Thank you for this valuable comment. We have changed in the text to incorporate the difference between DNA (a documented impact on couple fertility) and the other organelles (do not yet have a documented impact on couple fertility). By rewriting the sentence to: “A diagnosis of male infertility rarely includes an assessment of internal sperm components such as DNA, which is well documented to have an impact on infertility, or other components such as RNA and centrioles, which are beginning to be adopted.

  1. Line 21: “there is no treatment available”. It is not really true. For sperm DNA framentation, a therapy with FSH has been suggested (see and add as reference Simoni et al., Hum Reprod. 2016 Sep;31(9):1960-9.), as well as with antioxidants (even if debated), and some technologies, such as MACS can be used to select the “best” spermatozoa (even id debated).

Answer: Thank you for your suggestion. We rephrased that sentence to: “Assessment of these internal components is not typically included in current diagnostic testing because there are limited available treatments. Additionally We have added the specifics to the main body of the text in section 7, DNA Fragmentation.

  1. Introduction: see the comments above.

Answer: We have introduced the above points in the introduction

  1. Line 51. I do not agree with this point of view “In this paper, we suggest that further research of the internal components of the sperm is necessary to distribute the burden of infertility between the partners”. The aim of this study should be not to “distribute the burden”. Our aim as researchers is to find out new diagnostic and therapeutic options in order to increase our knowledge on male infertility and find new teratments to optimize natural and ART fecundity.

Answer: Thank you for your comment. We incorporated this language  as follows: “In this paper, we suggest that basic and translational research of sperm biology is needed to increase our knowledge of male infertility and find new treatments to optimize natural conception and ART outcomes.” Additionally, we deleted: “Doing so will redistribute the burden of infertility between the partners.” 

  1. Accordingly, please rephrase also “Identifying male components would shift the emphasis from treating women for male infertility to treating men for male infertility”.

Answer: We changed this part of the paper extensively to state “The contribution of men and women to reproduction is uneven. While men provide the sperm and women provide the egg, women must also carry and deliver the pregnancy. Therefore, it is a biological inevitability that women are physically burdened more than men by fertility treatments. However, a woman undergoing extensive fertility treatments for male infertility is not a biological inevitability. Instead, we argue that deficiencies in our understanding of the biology of male fertility and the common practice of solely employing traditional semen analysis to diagnose male infertility have led to an uneven distribution of treatment. This situation is largely due to the success and availability of assisted reproductive technology (ART), which disproportionately burdens women. The perception that ART has "solved" male infertility has hindered male infertility research, development of novel diagnostics, and advancement of treatments that would either improve treatment options or help men achieve natural conception [1]”

  1. Line 61. “the need for advanced and experimental sperm testing and male treatments to male infertility to alleviate this treatment inequality”. Again, this comment is dealing with sex inequality, but this is not the point (see the general comment and comment #16). Please rephrase.

Answer: Thank you for your suggestion. We changed the sentence to “Subsequently, we explain how the deficiencies in traditional semen analysis additionally burdens women. Furthermore, we claim that there are undiagnosed and potentially treatable sperm defects that could be addressed by exploring advanced and experimental sperm testing.” 

Main text

  1. Line 70. Please update reference 1 (see Lotti and Maggi. Nat Rev Urol. 2018 May;15(5):287-307)

Answer: Thank you, the reference has been added. 

  1. Line 74, Figure 1. Please report some other important correctable causes of male infertility, such as hypogonadotropic hypogonadism (treatment with FSH and/or hCG) (add a reference), oligo- and/or astheno and /or teratospermia (add a reference), midline prostatic cyst and obstructive infertility.

Answer: Thank you for this comment, the focus of figure 1 is male factors where the treatment is biased against women. Figure 1 only shows a few examples of potential diagnoses and treatments that are biased against women.

  1. Line 78. “semen analysis semen analysis” correct this mistake.

Answer: Thx - changed

  1. Line 80. Be very careful with this comment “For example, 40% of men from couples with recurrent pregnancy loss had normal sperm density and motility, but had abnormal sperm aneuploidy”. It seems that 40% of cases of miscarriages depend on male sperm aneuploidy, but it is not the case. The demonstrated male contribution to miscarriages is very low.

Answer: Thank you for your comment, we have modified our text: “. For example, it was reported in several studies that 40% of men from couples with recurrent pregnancy loss had normal sperm density and motility, but had abnormal sperm aneuploidy [17]. While the role of sperm fragmentation is controversial [18], there are examples where, even with normal bulk semen parameters, men can have sperm with abnormal components”  

  1. Line 97. “she may still feel responsible for being unable to conceive”. Please don’t be so definitive. There is a large literature about male negative feeling related to infertility (see and add as references Bechoua et al., Andrology 2016:4;395-403, Lotti and Maggi. Nat Rev Urol. 2018 May;15(5):287-307; Lotti et al., Hum Reprod. 2016 Dec;31(12):2668-2680.)

Answer: Thank you for this comment, we have decided to combine the discussion of the emotional aspects of infertility on men and women. Suggested references are included in the sentence: “Both partners can support each other through the stress they feel when each understands the conditions that confound their inability to conceive”

  1. Line 157. “One barrier to men seeking infertility treatment is the lack of specialists (andrologists)”. I do not agree with this comment. Have andrologists in Europe or United States a long waiting list? I don’t think so. The great barrier for men to seek infertility treatment is that most of the infertility centers do not request or include an andrological evaluation. This is the problem: this mentality could be changed. And this mentality is mainly derived by the fact that often infertility centers are guided by gynecologists only, with no andrologists or urologiststs in their team. So far, infertility centers tend to “push” the couple to ART, often avoiding the male infertiliy investigation and research fo natural pregnancy or pregnancy with IUI. Maybe, sometimes, also economic reasons underline these decisions. Several cases where natural pregnacy could be obtained treating the male partner or with female antiestrogens treatment, and IUI are often shifted to ART with the information (and sometimes the excuse) that that techniques will allow a higher success probability. Obviously, a different scenario is when the female partner is old, for example around 40 years old, when ART is necessary, especially after longlasting infertility, to avoid a repentine reduction of ovocite quality in the next few years. However, after female 43-45 years old, ART or natural sexual intercorses have similar chances to get pregnancy.

Answer: Thank you for this suggestion. Please see our changes to the sentence: “One barrier to men seeking infertility treatment is that male specialists (andrologists) are not usually part of the infertility treatment team. However, even if this were to change, there would be an insufficient number of andrologists.” We respectfully disagree in part with the  reviewer on this point. In much of the country it is not possible to find a reproductive urologist. They may not have long waiting lists, but if they are not physically there it is difficult to consult with them.

  1. Regarding DNA fragmentation, please add a paragraph on brighter and dimmer subpopulations (see and add as references Marchiani et al., Andrology. 2014 May;2(3):394-401; Muratori M, Fertil Steril. 2015 Sep;104(3):582-90.e4.) Please add also a discussion about MACS and similar technologies to select the “best” spermatozoa, and evidences for their use in clinical practice.

Answer: Thank you for this suggestion, we have added: “If the DNA fragmentation index is elevated, there are several potential treatments. These treatments include follicle-stimulating hormone (FSH) [1], magnetic assisted cell sorting (MACS) that may help select sperm with high-quality DNA [61], and retrieval of testicular sperm (known as TESE/TESA) for use with ICSI [62,63]. The diagnosis and treatments for DNA fragmentation have not been fully validated using randomized control trials. Future investigations with demonstrated sensitivity and specificity are required.”

  1. Please add a paragraph on physical (for example sexual) and psychological problems of the male partner due to infertility, and add a brief discussion on male infertility as a mirror of a poor general health (see and add as reference “Lotti and Maggi. Nat Rev Urol. 2018 May;15(5):287-307.”) Hence, underline the necessity to increase the evaluation of the male partner in infertility clinic, not only for fertility outcomes but also for investigating and treat general, sexual and psychological problems.

Answer: Please see response to comment 15.

  1. Please add a paragraph considering the utility and increasing reseasch of male genital tract ultrasound for male infertility (see Lotti and Maggi, Hum Reprod Update. 2015 Jan-Feb;21(1):56-83).

Answer: Thank you for this important addition.  We have introduced this point at the end of our introduction. Because this paper focuses on increasing sperm research, we added: “Of note, in this manuscript, we focus on the sperm’s role in male infertility; however, similar arguments can be made about other male elements. For example, the male genital tract [2], seminal fluids [reviewed in 3,4], hormonal regulation [5,6], non-germ line testicular cells [7,8], and his genetic constitution can all contribute to male infertility [9].” 

  1. Please add a paragraph considering the utility and increasing reseasch of genetics on male infertility (see Krausz et al., Expert Rev Mol Diagn. 2018 Apr;18(4):331-346).

Answer: Thank you for this important addition. We added it together with the response to comment 19.

  1. The manuscript should be reduced in some redundants parts.

Answer: Thank you for this comment, we eliminated section 3 that included many redundancies and reduced redundancies in the abstract.

  1. A more “scientific” English or American language is required.

Answer: While writing this paper, we have made a conscious decision to use more accessible language so it can be read by students and non-specialists.

Reviewer 2 Report

This is a well-written and clearly organized paper that discusses the often neglected topic of men’s infertility. The authors do a nice job of explaining how the lack of assisted reproductive technologies geared towards men’s bodies means that women bear most of the physical, and often emotional, burdens of infertility treatment.

Specific comments

  • Page 4, lines 117 – 123: in addition to listing the side effects of hormonal medication, it may be worth pointing out that the procedure of IUI can also be uncomfortable for women. Furthermore, IUI is invasive, which may be psychologically difficult for some women, especially women with a history of sexual assault. While providing a semen sample is not physically invasive, it can be emotionally stressful for some men.
  • Page 4, line 124: many in the field would label IUI as a form of ART because it is not something people can do without medical assistance. For the sake of clarification, it may be helpful to explain that there are various types of procedures that fall under the umbrella of ART, including IUI, IVF, ICSI, preimplantation genetic screening, donor gametes, and gestational surrogates. Furthermore, it may be useful to note that one doesn’t necessarily have to employ all of them (e.g. a couple can have IVF without ICSI), but that some procedures require the use of other procedures (e.g. ICSI can only be performed with IVF).
  • Page 5, lines 155 – 157: minor editorial note that it may seem redundant to list “knowledge” and then include “compounded with our lack of knowledge” in the following sentence: “There are multiple epidemiologic, geographic, knowledge, financial, socioeconomic, and policy-based barriers that, compounded with our lack of knowledge, make it challenging for men to obtain high-quality infertility care.” Also, all the words listed in this sentence are adjectives except the word “knowledge.” For the sake of consistency, you could replace the word “knowledge” here with the word “epistemic.”
  • Page 5, line 169: typo: change “reduce” to “reduced.”
  • Page 6, line 202: spell out the acronym MOSI.
  • Page 7, lines 268 – 272: although expensive, fertility preservation is an option for woman with low ovarian reserve and this would give the couple more time then needing to rush to use a sperm donor immediately.
  • Page 7, line 276: is the Michigan Action Plan for Fertility and Assisted Reproductive Technology (ART) well-known enough that saying that they believe examining male infertility is a top project will carry significant weight without further support?

Author Response

Review 2

This is a well-written and clearly organized paper that discusses the often neglected topic of men’s infertility. The authors do a nice job of explaining how the lack of assisted reproductive technologies geared towards men’s bodies means that women bear most of the physical, and often emotional, burdens of infertility treatment.

Answer: We would like to thank  the reviewer for recognising the contribution of the paper and for the valuable comments. We believe these comments help us to straighten the paper. 

Specific comments

  • Page 4, lines 117 – 123: in addition to listing the side effects of hormonal medication, it may be worth pointing out that the procedure of IUI can also be uncomfortable for women. Furthermore, IUI is invasive, which may be psychologically difficult for some women, especially women with a history of sexual assault. While providing a semen sample is not physically invasive, it can be emotionally stressful for some men.

Answer: Thank you for this addition.

We edited and added the following sentence: “Additionally, IUI is relatively non-invasive, but is physically uncomfortable.” Please note that this paper is not talking about the male discomfort, but focuses on female discomfort due to male infertility

  • Page 4, line 124: many in the field would label IUI as a form of ART because it is not something people can do without medical assistance. For the sake of clarification, it may be helpful to explain that there are various types of procedures that fall under the umbrella of ART, including IUI, IVF, ICSI, preimplantation genetic screening, donor gametes, and gestational surrogates. Furthermore, it may be useful to note that one doesn’t necessarily have to employ all of them (e.g. a couple can have IVF without ICSI), but that some procedures require the use of other procedures (e.g. ICSI can only be performed with IVF).

Answer: Thank you very much for this comment, even though ART can be defined in many ways we have added the broader suggestion made by the reviewer. Because in this paper we focus on male infertility’s effect on women we do not speak about preimplantation genetic screening, donor gametes, and gestational surrogates.

  • Page 5, lines 155 – 157: minor editorial note that it may seem redundant to list “knowledge” and then include “compounded with our lack of knowledge” in the following sentence: “There are multiple epidemiologic, geographic, knowledge, financial, socioeconomic, and policy-based barriers that, compounded with our lack of knowledge, make it challenging for men to obtain high-quality infertility care.” Also, all the words listed in this sentence are adjectives except the word “knowledge.” For the sake of consistency, you could replace the word “knowledge” here with the word “epistemic.”

Answer: Thx - changed

  • Page 5, line 169: typo: change “reduce” to “reduced.”

Answer: Thx - changed

  • Page 6, line 202: spell out the acronym MOSI.

Answer: Thx - added

  • Page 7, lines 268 – 272: although expensive, fertility preservation is an option for woman with low ovarian reserve and this would give the couple more time then needing to rush to use a sperm donor immediately.

Answer: Thank you for the comment, we have introduced this: “Another option is fertility preservation where the woman freezes her eggs to give clinicians more time to correct the male infertility.”

  • Page 7, line 276: is the Michigan Action Plan for Fertility and Assisted Reproductive Technology (ART) well-known enough that saying that they believe examining male infertility is a top project will carry significant weight without further support?

Answer: The Michigan Action Plan for Fertility and Assisted Reproductive Technology (ART) was created by the well-known state department of health. 

Reviewer 3 Report

Thank-you for the opportunity to review this paper which is a review and discussion piece regarding the lack of focus on the male in the management and treatment of infertility. The paper brings attention to an interesting and overlooked aspect of infertility – the excess burden carried by the female partner. However, I feel that some aspects lack clear and important elaboration and I have the following comments on the manuscript.

  1. I wonder if the authors might like to reconsider their language regarding ART as ‘treating the woman’ per se (e.g. lines 102-104). Couples might prefer to be thought of as having IVF together; the couple is treated with ART –sperm is involved after all. Referring to ART as treating the woman might be viewed as discounting the male’s involvement in the process.
  2. Might it be a biological inevitability that women are burdened more so than men by fertility treatments, in the same way they are burdened more in natural conceptions by carrying the pregnancy and childbirth? Women cannot eject oocytes out of their body easily -but a man can ejaculate. Only women have the organ to carry a pregnancy – the uterus, and embryo transfer therefore will always involve the women. It may be worth mentioning this and that the aim of any research into male fertility is not to achieve equity here – but to reduce the burden for women (or even both parties) wherever possible.
  3. Timed intercourse is mentioned (lines 114) but with little detail about what the burden might be exactly. Taking temperature or ovulation test kits? It might also be said that months and years of timed intercourse may indeed burden the male, who needs to ‘perform’ on demand, so to speak. There appears to be plenty of research on this topic e.g.: https://doi.org/10.1111/j.1464-410X.2012.11577.x
  4. I think there is a missing link in the narrative that treatments that focus on men might improve naturally fertility and therefore reduce the need for IVF/ICSI or for multiple IVF/ICSI – this is not quite spelt out. For example, consider lines 138-140 “It appears, that the successful use of the above-mentioned treatments has hindered the research and development of treatment options that focus on men” – it could be said that new techniques such as IMSI and PICSI do focus on men – they are measures for sperm selection – however these do not necessarily reduce the burden for the women who is still undergoing egg pick up etc. It is treatment options that focus on improving the man’s natural potential for pregnancy which we need, and which improve IVF success such that repeat IVF cycles are not required. I think it might help to labour this point.
  5. An underlying assumption of this view is that there are undiagnosed and treatable defects of sperm. There is a smattering of evidence provided regarding defects (e.g. lines 133-135 and lines 80-81), however it is not condensed anywhere to help in framing the need for these tests. I wonder if a subheading on this point would be helpful. For example, line 211 states “Because we now realize the importance of sperm health” – do we? Where is the evidence for this?
  6. It is stated that if the DNA frag index is high, the man can proceed with a TESA and ICSI. Is there evidence from RCTs that this approach actually leads to a higher conception rate? The cited references are not RCTs. I would suggest the authors remove the statement that these are validated measures, stating that any test should be proven to be reach both sufficient levels of sensitivity and specificity, and that any treatments suggested on the basis of these tests need to be proven safe and effective. Critics might suggest DNA fragmentation testing is currently a diagnostic IVF ‘add-on’ - proven to do nothing but cost patients more money… However, there may be an argument that a diagnostic test is helpful even if there are no obvious treatment strategies, if the test itself is predictive of the chance of success from IVF. This information can then be used to counsel couples in their decision whether to proceed with possibly futile treatment, as the authors describe in lines 267-272.
  7. The authors may be aware that an international Priority Setting Partnership for infertility identified the following as a top research question for male infertility “Are sperm tests other than the WHO parameters useful in evaluating male fertility?” https://www.focusonreproduction.eu/article/ESHRE-Meetings-research-gaps-19

I have the following very minor comments/corrections

  1. Lines 55-56 don’t flow very well to the reader
  2. Line 78 repeats ‘semen analysis’ twice
  3. I don’t understand this sentence, lines 78-80 “This is because semen analysis semen analysis does not provide information on the state of the sperm contents and it is characteristic of unexplained infertility”
  4. Line 169 reduce should be reduced

Author Response

Review 3

Thank-you for the opportunity to review this paper which is a review and discussion piece regarding the lack of focus on the male in the management and treatment of infertility. The paper brings attention to an interesting and overlooked aspect of infertility – the excess burden carried by the female partner. However, I feel that some aspects lack clear and important elaboration and I have the following comments on the manuscript.

Answer: Thank you for recognizing the importance of excess burden women experience due to male infertility. We would like to further thank you for all the comments that have helped us improve the paper.

  1. I wonder if the authors might like to reconsider their language regarding ART as ‘treating the woman’ per se (e.g. lines 102-104). Couples might prefer to be thought of as having IVF together; the couple is treated with ART –sperm is involved after all. Referring to ART as treating the woman might be viewed as discounting the male’s involvement in the process.

Answer: Thank you for this note. We agree that ART treats the couple infertility. The issue is that as it is now, most of the burden (pain) falls on the side of the female. Accordingly, we change the text to: “For example, the woman may carry most of the treatment burden when the man has a low count, reduced motility, or has abnormal sperm morphology [19]. 

  1. Might it be a biological inevitability that women are burdened more so than men by fertility treatments, in the same way they are burdened more in natural conceptions by carrying the pregnancy and childbirth? Women cannot eject oocytes out of their body easily -but a man can ejaculate. Only women have the organ to carry a pregnancy – the uterus, and embryo transfer therefore will always involve the women. It may be worth mentioning this and that the aim of any research into male fertility is not to achieve equity here – but to reduce the burden for women (or even both parties) wherever possible.

Answer: Thank you for this note. We agree that men and women's reproductive biology is distinct in many aspects, such as pregnancy, and rely heavily on women's contribution. But why should a woman be the target of the treatment when a man has a problem in producing sperm? This type of infertility should be resolved by treating male infertility in the men's body or in vitro.

To address this comment we added the following to the text in the beginning of the paper: “The contribution of men and women to reproduction is uneven. While men provide the sperm and women provide the egg, women must also carry and deliver the pregnancy. Therefore, it is a biological inevitability that women are physically burdened more than men by fertility treatments. However, a woman undergoing extensive fertility treatments for male infertility is not a biological inevitability.”

  1. Timed intercourse is mentioned (lines 114) but with little detail about what the burden might be exactly. Taking temperature or ovulation test kits? It might also be said that months and years of timed intercourse may indeed burden the male, who needs to ‘perform’ on demand, so to speak. There appears to be plenty of research on this topic e.g.: https://doi.org/10.1111/j.1464-410X.2012.11577.x

Answer: Thank you for this comment, we added: “This treatment can be equally stressful on the couple because both partners need to perform on-demand [23].” 

I think there is a missing link in the narrative that treatments that focus on men might improve naturally fertility and therefore reduce the need for IVF/ICSI or for multiple IVF/ICSI – this is not quite spelt out. For example, consider lines 138-140 “It appears, that the successful use of the above-mentioned treatments has hindered the research and development of treatment options that focus on men” – it could be said that new techniques such as IMSI and PICSI do focus on men – they are measures for sperm selection – however these do not necessarily reduce the burden for the women who is still undergoing egg pick up etc. It is treatment options that focus on improving the man’s natural potential for pregnancy which we need, and which improve IVF success such that repeat IVF cycles are not required. I think it might help to labour this point.

Answer: Thank you very much for this comment that addressed this gap that was in the paper. We have now introduced throughout the paper the concept that the treatment goal is to improve the man’s fertility and enable natural conception. 

  1. An underlying assumption of this view is that there are undiagnosed and treatable defects of sperm. There is a smattering of evidence provided regarding defects (e.g. lines 133-135 and lines 80-81), however it is not condensed anywhere to help in framing the need for these tests. I wonder if a subheading on this point would be helpful. For example, line 211 states “Because we now realize the importance of sperm health” – do we? Where is the evidence for this?

Answer: Thank you very much for this great idea, we have added a section “There are undiagnosed and potentially treatable sperm defects” where we discuss this point. 

  1. It is stated that if the DNA frag index is high, the man can proceed with a TESA and ICSI. Is there evidence from RCTs that this approach actually leads to a higher conception rate? The cited references are not RCTs. I would suggest the authors remove the statement that these are validated measures, stating that any test should be proven to be reach both sufficient levels of sensitivity and specificity, and that any treatments suggested on the basis of these tests need to be proven safe and effective. Critics might suggest DNA fragmentation testing is currently a diagnostic IVF ‘add-on’ - proven to do nothing but cost patients more money… However, there may be an argument that a diagnostic test is helpful even if there are no obvious treatment strategies, if the test itself is predictive of the chance of success from IVF. This information can then be used to counsel couples in their decision whether to proceed with possibly futile treatment, as the authors describe in lines 267-272.

Answer: Thank you for this comment, we added: “The diagnosis and treatments for DNA fragmentation have not been fully validated using randomized control trials. Future investigations with demonstrated sensitivity and specificity are required.” 

We also added to section 9.

 A future diagnostic test may be helpful even if there are no obvious treatment strategies because it may predict whether the use of ART will be successful. This information could then be used to counsel couples in their decision on whether to proceed with possibly expensive and futile treatments.” 

“Identifying male infertility in infertile couples can have two benefits. In the short term, it can direct treatment efforts and get the sperm count to a range where IUI may be an option instead of IVF/ICSI. These efforts are beneficial because IUI is a less invasive and less expensive treatment than IVF/ICSI. In the long term, it can direct research to allow for the discoveries of treatments that improve male infertility to a point where it enables natural conception.”

  1. The authors may be aware that an international Priority Setting Partnership for infertility identified the following as a top research question for male infertility “Are sperm tests other than the WHO parameters useful in evaluating male fertility?” https://www.focusonreproduction.eu/article/ESHRE-Meetings-research-gaps-19

Answer: Thank you for this comment, we added it in our concluding remark: ““Are sperm tests other than the WHO parameters useful in evaluating male fertility?” is a top priority question according to An International Priority Setting Partnership for Infertility https://www.focusonreproduction.eu/article/ESHRE-Meetings-research-gaps-19.”

I have the following very minor comments/corrections

  1. Lines 55-56 don’t flow very well to the reader

Answer: Thx - we changed the sentence to: “This manuscript is intended to prompt discussion and investigation — it is not intended as a guideline — but is instead aimed to push the field of male infertility research forward.”

  1. Line 78 repeats ‘semen analysis’ twice

Answer: Thx - changed

  1. I don’t understand this sentence, lines 78-80 “This is because semen analysis semen analysis does not provide information on the state of the sperm contents and it is characteristic of unexplained infertility”

Answer: Thank you for noticing this mistake, we have corrected the sentence: “Additionally, it is important to note that men with a normal semen analysis merit additional diagnostics.”

  1. Line 169 reduce should be reduced

Answer: Thx - changed

Round 2

Reviewer 1 Report

I evaluated the manuscript cells-742897

In my opinion, the quality of the new version of the manuscript has improved. However, I have some minor comments to be amended before publication.

1.Line 52. “Therefore, it is a biological inevitability that women are physically 52 burdened more than men by fertility treatments”. Please rephrase ““Therefore, in infertile couples, it is a biological inevitability that women are physically burdened more than men by fertility treatments”.

2.Line 110. “For example, it was reported in several studies that 40% of men from couples with recurrent pregnancy loss had normal sperm density and motility, but had abnormal sperm aneuploidy [17]”. Again, are “several” studies reporting this percentage? I suggest to write that “some studies (cit) report that up to 40% of men…”

3.Line 204. “However, even if this were to change, there would be an insufficient number of andrologists”. Since the present manusript is a review, it should be underlined that “in USA” there would be an insufficient number of andrologists. For example, this is not the case of some Countries of Europe, where there are several andrologists (Italy, Spain) or urologists with andrological competence (Germany, Greece). Please report in which Countries/Continents of the world an insufficient number of andrologists could represent a problem, this could be a very interesting information.

4.Line 204.Consider also that in Germany several dermatologists acts as andrologists, while in Italy both endocrinologists and urologists can be andrologists (andrology is not a specialization in Italy), however endocrinologist-andrologists are better than urologist-andrologist because the former has a high “medical” formation, while the latter have a surgical formation. Please add thes concept to the manuscript.

Author Response

In my opinion, the quality of the new version of the manuscript has improved. However, I have some minor comments to be amended before publication.

Answer: We would like to thank the reviewer for recognizing that the new version of the manuscript has improved and his comments. We believe these comments help us to strengthen the paper.

1.Line 52. “Therefore, it is a biological inevitability that women are physically 52 burdened more than men by fertility treatments”. Please rephrase ““Therefore, in infertile couples, it is a biological inevitability that women are physically burdened more than men by fertility treatments”.

Answer: Thx – we made the change.

2.Line 110. “For example, it was reported in several studies that 40% of men from couples with recurrent pregnancy loss had normal sperm density and motility, but had abnormal sperm aneuploidy [17]”. Again, are “several” studies reporting this percentage? I suggest to write that “some studies (cit) report that up to 40% of men…”

Answer: Thx – we made the change.

3.Line 204. “However, even if this were to change, there would be an insufficient number of andrologists”. Since the present manusript is a review, it should be underlined that “in USA” there would be an insufficient number of andrologists. For example, this is not the case of some Countries of Europe, where there are several andrologists (Italy, Spain) or urologists with andrological competence (Germany, Greece). Please report in which Countries/Continents of the world an insufficient number of andrologists could represent a problem, this could be a very interesting information.

Answer: Thx – we changed the sentence to: “However, even if this were to change, there would be an insufficient number of andrologists in the United States, where there are only approximately 200 andrologists [38].”

4.Line 204.Consider also that in Germany several dermatologists acts as andrologists, while in Italy both endocrinologists and urologists can be andrologists (andrology is not a specialization in Italy), however endocrinologist-andrologists are better than urologist-andrologist because the former has a high “medical” formation, while the latter have a surgical formation. Please add thes concept to the manuscript.

Answer: Thx – we Added “For historical reasons, in some European countries, physicians such as dermatologists, and endocrinologists can act as andrologists [Glander, 2007 #5065].”